# SCformer: Segment Correlation Transformer for Long Sequence Time Series Forecasting

## Abstract

Long-term time series forecasting is widely used in real-world applications such as financial investment, electricity management and production planning. Recently, transformer-based models with strong sequence modeling ability have shown the potential in this task. However, most of these methods adopt point-wise dependencies discovery, whose complexity increases quadratically with the length of time series, which easily becomes intractable for long-term prediction. This paper proposes a new Transformer-based model called SCformer, which replaces the canonical self-attention with efficient segment correlation attention (SCAttention) mechanism. SCAttention divides time series into segments by the implicit series periodicity and utilizes correlations between segments to capture long short-term dependencies. Besides, we design a dual task that restores past series with the predicted future series to make SCformer more stable. Extensive experiments on several datasets in various fields demonstrate that our SCformer outperforms other Transformer-based methods and training with the additional dual task can enhance the generalization ability of the prediction model.

## 1 Introduction

Time series forecasting has always been a classic machine learning problem. It is widely used in various fields that are closely related to our lives, e.g., production planning, financial investment, traffic management, and electricity management. In many cases, we need to predict the future value of the time series for a long time, which involves long sequence time-series forecasting (LSTF) (Zhou et al., 2021). Classical mathematical forecasting models (Lütkepohl, 2005; Box et al., 2015; Cao & Tay, 2003; Roberts et al., 2013) rely on strong assumptions on the time series and can only handle simple linear relationships. For example, the ARIMA model (Box et al., 2015) requires the original time series to be stable at least after the differentiation operation. Gaussian Process (Roberts et al., 2013) utilizes all the features to make predictions and fails in high-dimensional data space.

Due to the ability of modeling long-term and nonlinear relationships for large-scale and complicated sequential data, deep learning models have achieved better performance than these classical methods. Existing time series forecasting methods based on deep learning can be divided into three categories, i.e., RNN-based models (Lai et al., 2018; Salinas et al., 2020; Qin et al., 2017; Song et al., 2018), TCN-based models (Borovykh et al., 2017; Sen et al., 2019) and Transformer-based models (Vaswani et al., 2017; Wu et al., 2020a). RNN-based methods suffer from the problem of gradient vanishing, gradient exploding, and lack of parallelism. TCN-based methods need deeper layers to achieve larger local receptive fields. For both categories of methods, signals must pass through a long path between two far-away temporal locations, hence the number of operations required to associate two elements increases with their temporal distance. Differently, transformer-based methods directly model the relationships between any element pairs and can better capture long-term dependencies, which is crucial for LSTF.

On the other hand, the self-attention mechanism of Transformer calculates all similarities between any element pairs. The computational and space complexities increase quadratically with the length of the time series. Directly applying Transformer to LSTF is not only inefficient but also difficult to capture truly effective attention. Recent works (Zhou et al., 2021; Li et al., 2019; Kitaev et al., 2019) explore different sparse attention mechanisms to suppress the contribution of irrelevant time steps and ease the computational pressure to a certain extent. These models still perform dot-product

attention to time steps individually and utilize the point-wise connections to capture temporal dependencies. However, for LSTF, there is a strong correlation between neighboring points. A single point may have limited influence on predicting the future. Autoformer (Wu et al., 2021) conducts the series-wise dependencies discovery by performing auto-correlation of the time series to the top-k time delayed series. All points in the whole delayed series must share the same weight for aggregating the prediction and complicated fast Fourier transforms are required for auto-correlation.

These methods perform the correlation either at the point level or at the overall series level, which not only require high computational redundancy to intensively tackle point pairs or perform time-frequency domain transformations, but also do not directly reflect the true dependencies within the time series. For instance, in traffic flow prediction, to predict the flow of the future period, the flows of a single previous time point and the shifted whole time series may have limited contribution. There may be stronger correlations at the segment level, e.g., the flow of the peak time period today should be more related to the flow of the peak time period yesterday than the flow of the off-peak time period any day and the flow of a period that has passed a long time ago.

In this paper, we propose a novel sparse attention mechanism called *segment correlation attention (SCAttention)*. Different from Auto-Correlation in Autoformer, SCAttention segments the time series based on implicit period and regards every segment as one unit to compute the correlation between segments. We extend the point-wise dot product in conventional transformer-based attention mechanism to the segment-wise correlation, where segments do not need to be transformed into the frequency domain. Segment-wise correlation not only reduces the amount of calculation since the number of segments is much smaller than the number of points, but also can be combined with other sparse attention mechanisms. We derive our prediction model for LSTF, namely *Segment Correlation Transformer (SCformer)*, via simply replacing the canonical attention in the original Transformer model with SCAttention.

A stable predictor should generate consistent predictions, i.e., for any given historical time series, if the predicted future series is reversed as the input, the output of the predictor should be consistent with the reverse of the historical time series. Motivated by this idea, we design a dual task as a regularization to train our SCformer model in order to achieve more robust predictions. Different from AST (Wu et al., 2020b) which introduces adversarial loss to alleviate the error accumulation with an additional discriminator, our dual task-based regularization term only needs to reverse the predicted time series as the input to SCformer without introducing additional network structures and parameters.

The main contributions of this paper are summarized as follows:

- We propose a segment correlation attention (SCAttention) mechanism to replace the canonical point-wise attention mechanism. In addition to increasing efficiency, SCAttention also extracts more relevant information from the sequence. Besides, it can be easily combined with other sparse attention mechanisms for further improvement.

- We design a dual task as a regularization term in the training phase. The dual reverse prediction task restores the past by the future. By encouraging the forward and backward predictions to be consistent, the learned SCformer generates more stable predictions. Ablation studies demonstrate the effect of the dual task.

- Extensive experiments show that our SCformer model achieves better forecasting performance than other Transformer-based prediction models.

## 2 RELATED WORK

**Time Series Forecasting.** Early works on the TSF problem are based on classical mathematical models such as vector autoregression (VAR) (Lütkepohl, 2005) and auto regressive intergrated moving average (ARIMA) (Box et al., 2015). Support vector regression (SVR) (Cao & Tay, 2003) introduces a traditional machine learning method to regress the future. Gaussian Process (Roberts et al., 2013) predicts the distribution of future values without assuming any certain form of the prediction function. However, all these classical models can not handle complicated and unknown data distributions or high-dimensional data.

With the development of deep learning, neural networks have shown stronger modeling ability than classical models. Recurrent Neural Network (RNN) (Connor et al., 1992) and Temporal Convolution Network (TCN) (Oord et al., 2016; Bai et al., 2018) are two common types of deep models for modeling sequence data. LSTNet (Lai et al., 2018) combines convolutional layers and recurrent layers to capture both long-term and short-term dependencies. DeepAR (Salinas et al., 2020) predicts the parameters of future distributions in an auto-regressive fashion. There are also some works (Qin et al., 2017; Song et al., 2018) that introduce additional attention mechanism to RNN to achieve better performance in forecasting. However, RNN-based models suffer from the gradient vanishing and gradient exploding problem. Popular variants of RNN such as LSTM (Hochreiter & Schmidhuber, 1997) and GRU (Chung et al., 2014) can not solve this problem fundamentally. The lack of parallelizability is another main limitation of RNN-based models. Benefiting from the good parallelism of convolution operations, TCN-based models have also been used in time series tasks and achieved good results (Borovykh et al., 2017; Sen et al., 2019). Both RNN-based and TCN-based models do not explicitly model the dependencies between two far-away temporal locations, but the information exchange between them must go through a long path.

**Transfomer-based models.** Transformer (Vaswani et al., 2017) was originally proposed as a sequence-to-sequence model in natural language processing (NLP) to deal with machine translation. Due to its powerful and flexible modeling capabilities, it has even been widely applied in processing non-sequential data such as images in computer vision (CV) tasks (Dosovitskiy et al., 2020; Carion et al., 2020). Because of its huge success in NLP and CV, efforts have been made to adopt Transformer to solve the TSF problem (Li et al., 2019; Zhou et al., 2021; Wu et al., 2021; 2020b;a). Self-attention (Vaswani et al., 2017) plays an important role in Transformer for explicitly discovering the dependencies between any element pairs, but both the time and space complexities increase quadratically with the length of the sequence, which limits the application of Transformer in LSTF (Zhou et al., 2021).

Therefore, various spare self-attention mechanisms for improving the efficiency of Transformer have been proposed in recent years. Logfomer (Li et al., 2019) proposes LogSparse self-attention which selects elements in exponentially increasing intervals to break the memory bottleneck. Informer (Zhou et al., 2021) defines a sparsity measurement for queries and selects dominant queries based on this measurement to obtain ProbSparse self-attention. Reformer (Kitaev et al., 2019) reduces the time and memory complexity by locally sensitive hashing self-attention.

Most works use point-wise dot product to compute attention score, and differ in the way of selecting point pairs. AutoFomer (Wu et al., 2021) develops an Auto-Correlation mechanism to replace self-attention, which utilizes series-wise correlation instead of point-wise dot product. In this work, we introduce a new segment correlation attention mechanism to explore the context information within neighboring points and capture the segment-wise correlation in the sequence. Our method differs from the Auto-Correlation mechanism (Wu et al., 2021) in the way of correlation computation and aggregation. Instead of the complicated Fast Fourier Transforms calculation in Auto-Correlation, we directly segment the time series based on implicit period and compute the correlation between segments. Besides, we aggregate the segments by inter-segmentation correlation.

AST (Wu et al., 2020b) regards the predictor as a generator and utilizes adversarial training as the regularization for the sequence-level forecasting of time series. The adversarial training requires another discrimination network to distinguish the predicted sequences and the ground-truth sequences. Differently, to make the forecasting process more stable, we design a dual task for regularization without introducing additional parameters.

# 3 METHOD

## 3.1 PROBLEM DEFINITION

We follow the comprehensive problem definition and setting about multi-horizon forecasting provided in (Lim et al., 2021). Typically, given the previous time series $\mathbf{X}_{1:t_0} = \{\mathbf{x}_1, \mathbf{x}_2, \ldots, \mathbf{x}_{t_0}\}$, where $\mathbf{x}_t \in \mathbb{R}^{d_x}$ and $d_x$ is the dimensionality of the variable, we aim to predict the future values $\mathbf{Y}_{t_0+1:t_0+\tau} = \{\mathbf{y}_{t_0+1}, \mathbf{y}_{t_0+2}, \ldots, \mathbf{y}_{t_0+\tau}\}$, where $\mathbf{y}_t \in \mathbb{R}^{d_y}$ is the prediction at every time step $t$ and

$d_y$ is the dimension of the output variable. The prediction model $f$ can be formulated as:

$$\hat{\mathbf{Y}}_{t_0+1:t_0+\tau} = f(\mathbf{X}_{1:t_0}; \Omega), \tag{1}$$

where $\hat{\mathbf{Y}}_{t_0+1:t_0+\tau}$ is the predicted time series and $\Omega$ is the learnable parameters of the model. For LSTF, the prediction range $\tau$, i.e., the future time duration to be predicted, is longer. The problem can be categorized into two types based on whether the dimension of the output variable $d_y$ is larger than one: univariate LSTF and multivariate LSTF.

## 3.2 THE SCFORMER MODEL

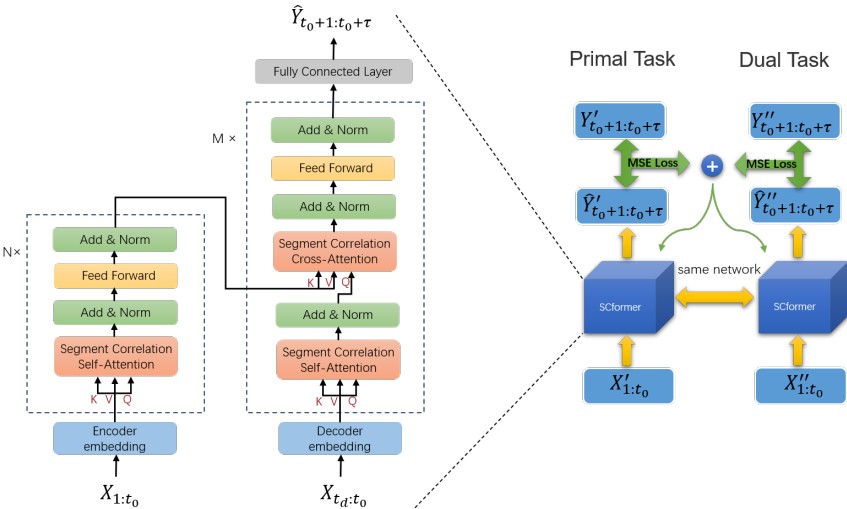

Figure 1: The overall framework of the SCformer model. Left: the architecture. Right: the dual task in the training stage.

As shown in Figure 1, the overall architecture of SCformer is similar to the vanilla Transformer. The original time series $\mathbf{X}_{1:t_0}$ is fed into the encoder and decoder through an embedding layer. We replace dot-product self-attention in Transformer with SCAttention. Besides, a fully connected layer is added to convert the output of the decoder into prediction values $\hat{\mathbf{Y}}_{t_0+1:t_0+\tau}$.

**Embedding.** Some prediction models utilize various time-dependent features (e.g., a set of dummy variables like hour-of-the-day, day-of-the-week, etc) and time-independent features(e.g., location of the store, car lane id, etc) to help prediction. But in this paper, we do not use these additional features as parts of inputs considering that useless noises are not conducive to prediction. Therefore, the inputs for our model only include values of the original time series. The input $\mathbf{X} \in \mathbb{R}^{L \times d}$ is transformed into $\mathbf{H} \in \mathbb{R}^{L \times d_{model}}$, followed by the commonly used absolute sinusoidal position encoding(Vaswani et al., 2017):

$$\mathbf{H} = \text{Embed}(\mathbf{X}) = \mathbf{X}\boldsymbol{W} + \mathbf{E}_{pos}, \tag{2}$$

where $\boldsymbol{W} \in \mathbb{R}^{d \times d_{model}}$ is a learnable projection matrix and $\mathbf{E}_{pos}$ is exactly the same as the positional encoding in Vaswani et al. (2017). The inputs of both encoder and decoder are processed by the above way.

**Encoder.** The encoder consists of $N$ identical layers, where each layer consists of a SCAttention module and a feed forward network (FFN) each followed by layer normalization with residual connections. The inputs of encoder are the past $t_0$ time steps $\mathbf{X}_{1:t_0} \in \mathbb{R}^{t_0 \times d_x}$. Then the encoder embedding layer converts $\mathbf{X}_{1:t_0}$ into $\mathbf{H}_{en}^0 \in \mathbb{R}^{t_0 \times d_{model}}$. Subsequently, for the $l$-th encoder layer, the input is the hidden representation of the the $l-1$-th layer and the output can be calculated as:

$$\begin{aligned}\mathbf{H}_{en}^{l,1} &= \text{LayerNorm}(\text{SCAttention}(\mathbf{H}_{en}^{l-1}) + \mathbf{H}_{en}^{l-1}) \\ \mathbf{H}_{en}^{l} &= \text{LayerNorm}(\text{FFN}(\mathbf{H}_{en}^{l,1}) + \mathbf{H}_{en}^{l,1}).\end{aligned} \tag{3}$$

$\mathbf{H}_{en}^N$ denotes the output of the encoder which will be fed into the cross-attention module in each decoder layer.

**Decoder.** The decoder consists of $M$ identical layers. Different from the encoder, there is an additional multi-head cross-attention layer where key and value matrices are transformed by the output of the encoder in each decoder layer. The inputs $\mathbf{X}_{t_d:t_0}$ of the decoder are the latter parts of the inputs of the encoder, where $t_d$ is the start time for the inputs of the decoder and $1 \le t_d \le t_0$. After the decoder embedding layer, $\mathbf{H}_{de}^0 \in \mathbb{R}^{(t_0-t_d) \times d_{model}}$ will be fed into the first decoder layer. For the $l$-th decoder layer, the input is the concatenation of the hidden representation of the $l-1$-th decoder layer and the output of the encoder. Its output can be calculated as:

$$\mathbf{H}_{de}^{l,1} = \text{LayerNorm}(\text{SCAttention}(\mathbf{H}_{de}^{l-1}) + \mathbf{H}_{de}^{l-1})$$
$$\mathbf{H}_{de}^{l,2} = \text{LayerNorm}(\text{SCAttention}(\mathbf{H}_{de}^{l,1}, \mathbf{H}_{en}^N) + \mathbf{H}_{de}^{l,1}) \qquad (4)$$
$$\mathbf{H}_{de}^l = \text{LayerNorm}(\text{FFN}(\mathbf{H}_{de}^{l,2}) + \mathbf{H}_{de}^{l,2}).$$

An additional fully connected layer takes $\mathbf{H}_{de}^l$, the output of $M$-th decoder layer, as input, and it outputs final prediction values $\hat{\mathbf{Y}}_{t_0+1:t_0+\tau} \in \mathbb{R}^{\tau \times d_y}$:

$$\hat{\mathbf{Y}}_{t_0+1:t_0+\tau} = \text{FC}(\mathbf{H}_{de}^M), \qquad (5)$$

where FC not only includes fully connected projection but also includes the flatten and resize operations to get prediction values with dimension $d_y$.

## 3.3 Segment Correlation Attention

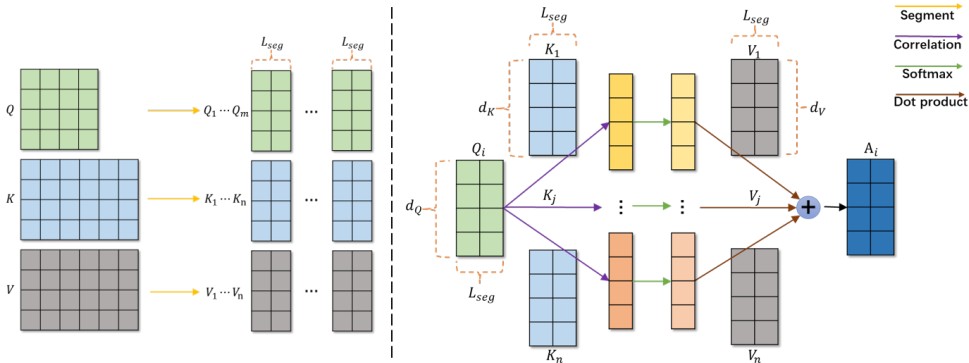

Figure 2: Simplified schematic of SCAttention. Segmentation operation (left) and correlation attention calculation for each query segment $\mathbf{Q}_i$ (right).

SCAttention is the key module in our SCformer, which performs segment-wise attention instead of point-wise attention. We denote the input of each SCAttention module as $\mathbf{H} \in \mathbb{R}^{L \times d_{model}}$[1], where $L$ and $d_{model}$ are the length of the time series and the dimension of variables respectively. Formally, for the single head situation, the input series $\mathbf{H}$ will be projected by three projection matrices to obtain the query, key and value, i.e., $\mathbf{Q} = \mathbf{H}\mathbf{W}_Q, \mathbf{K} = \mathbf{H}\mathbf{W}_K, \mathbf{V} = \mathbf{H}\mathbf{W}_V$, where $\mathbf{W}_Q, \mathbf{W}_K, \mathbf{W}_V \in \mathbb{R}^{d_{model} \times d}$, respectively. Then all the $\mathbf{Q}$, $\mathbf{K}$ and $\mathbf{V}$ are segmented into several segments having the same length $L_{seg}$:

$$\text{SEG}(\mathbf{Q}, L_{seg}) = \{\mathbf{Q}_1, \mathbf{Q}_2, \dots, \mathbf{Q}_m\}, \mathbf{Q}_i \in \mathbb{R}^{L_{seg} \times d}$$
$$\text{SEG}(\mathbf{K}, L_{seg}) = \{\mathbf{K}_1, \mathbf{K}_2, \dots, \mathbf{K}_n\}, \mathbf{K}_i \in \mathbb{R}^{L_{seg} \times d} \qquad (6)$$
$$\text{SEG}(\mathbf{V}, L_{seg}) = \{\mathbf{V}_1, \mathbf{V}_2, \dots, \mathbf{V}_n\}, \mathbf{V}_i \in \mathbb{R}^{L_{seg} \times d},$$

where $m = n = L//L_{seg}$ if query and key come from the same input. $\text{SEG}(\mathbf{X}, l)$ represents the segmentation operation that divides $\mathbf{X}$ into several segments with the same length $l$. Generally, we

---

[1]For convenience, we omit the layer index and the subscript which identifies the encoder or decoder.

set the segment length $L_{seg}$ to the implicit period of the time series. However, since the number of segments is more than 1, it is also an appropriate choice to set $L_{seg}$ to a factor of the period. More exploration on $L_{seg}$ is described in Section 4.3. The segmentation operation is shown in Figure 2.

The correlation vector $\mathbf{c}_j^i$ between any pair of query segment $\mathbf{Q}_i$ and key segment $\mathbf{K}_j$ can be computed by the Correlation function:

$$\mathbf{c}_j^i = \text{Correlation}(\mathbf{Q}_i, \mathbf{K}_j), \tag{7}$$

where the Correlation function is implemented by computing the reduce-sum of element-wise product between two matrices with same size. For each query segment $\mathbf{Q}_i$, all the correlation vectors will be normalized by the Softmax function to get the weight vector $\{\hat{\mathbf{c}}_j^i \mid j = 1, \ldots, n\}$:

$$\hat{\mathbf{c}}_1^i, \ldots, \hat{\mathbf{c}}_n^i = \text{Softmax}(\mathbf{c}_1^i, \ldots, \mathbf{c}_n^i). \tag{8}$$

Considering that the importance of different dimensions differs, we use weight vectors to aggregate all the value segments instead of scalar weights. Specifically, $\mathbf{A}_i$ is the weighted sum of all the value segments $\{\mathbf{V}_j \mid j = 1, \ldots, n\}$, as shown in Figure 2:

$$\mathbf{A}_i = \sum_{j=1}^n \hat{\mathbf{c}}_j^i \odot \mathbf{V}_j, \tag{9}$$

where $\odot$ is the dot product operator with broadcasting between one matrix and one vector. Lastly, output of SCAttention layer can be obtained by concatenating all the $\mathbf{A}_i$ along length dimension:

$$\text{SCAttention}(\mathbf{Q}, \mathbf{K}, \mathbf{V}) = \text{Concat}(\mathbf{A}_1, \ldots, \mathbf{A}_m). \tag{10}$$

The output of multi-head SCAttention can be computed by concatenating and projecting the outputs of all heads described above. We omit the formulation since it is similar to the canonical multi-head attention (Vaswani et al., 2017).

As a result, all time steps in a segment share the same aggregation weight vector, as opposed to dot-product attention, where each time step has its own aggregation scalar weight. This segment-wise attention reduces redundant point-wise calculations so that both time and memory efficiency have been improved, i.e., the complexity is reduced to $\mathcal{O}(L^2/L_{seg})$.

## 3.4 DUAL TASK

Many real time series have shown certain seasonal periodicity (Hyndman & Athanasopoulos, 2018), and it should have a steady upward trend and a steady downward trend. If a prediction model is powerful enough to use past sequences to predict the future, it should be able to use future sequences to restore the past in turn. On the contrary, if the past sequences recovered by the predictor differ greatly from the inputs, the predictor is not stable and the predictions are not reliable. Motivated by this idea, we design a dual task as a regularization to train our SCformer model in order to achieve more robust prediction performance.

Dual task training is shown in the right part of Figure 1. The primal task, i.e., the forward prediction task, is described in (1). We choose the MSE loss function on model prediction w.r.t the ground truth. The reverse prediction task aims to restore the past series according to the future data.

$\mathbf{X}_{1:t_0+\tau}$ and $\mathbf{Y}_{1:t_0+\tau}$ denote original time series and target values during the whole duration, respectively. For the forward prediction task, $\mathbf{X}'_{1:t_0} = \mathbf{X}_{1:t_0}$ is fed into the model to get $\hat{\mathbf{Y}}'_{t_0+1:t_0+\tau}$ and the ground truth is $\mathbf{Y}'_{t_0+1:t_0+\tau} = \mathbf{Y}_{t_0+1:t_0+\tau}$. Therefore, the loss of forward prediction task is:

$$\mathcal{L}_{forward} = \text{MSE}(\hat{\mathbf{Y}}'_{t_0+1:t_0+\tau}, \mathbf{Y}'_{t_0+1:t_0+\tau}). \tag{11}$$

For the reverse prediction task, the input of the model is $\mathbf{X}''_{1:t_0} = \mathbf{X}_{t_0+\tau:1+\tau:-1}$, where $-1$ in subscript indicates that the index is decreasing at continuous interval. $\hat{\mathbf{Y}}''_{t_0+1:t_0+\tau}$ and $\mathbf{Y}''_{t_0+1:t_0+\tau} = \mathbf{Y}_{\tau:1:-1}$ denote the output and the ground truth respectively, and the loss of dual task can be formulated as:

$$\mathcal{L}_{reverse} = \text{MSE}(\hat{\mathbf{Y}}''_{t_0+1:t_0+\tau}, \mathbf{Y}''_{t_0+1:t_0+\tau}). \tag{12}$$

Therefore, the total loss function $\mathcal{L}$ can be defined by:

$$\mathcal{L} = \mathcal{L}_{forward} + \lambda \mathcal{L}_{reverse}, \tag{13}$$

where $\lambda$ is the hyper-parameter that balances losses of two tasks.

## 4 EXPERIMENTS

### 4.1 EXPERIMENTAL SETUP

**Datasets**   We conduct experiments on the following four datasets as in Zhou et al. (2021). (1) *ETT*[2] (Zhou et al., 2021) contains data related to electricity which is collected from two Chinese stations in two years. In order to explore the model's performance on data with different granularities, we use different sampling frequencies to get hourly data {*ETTh1, ETTh2*} and 15-minutes data *ETTm1*. The train/val/test contains 12/4/4 months of data. (2) *Weather* [3] collects hourly climatological data of about 1600 U.S. locations in 4 years, which contains 11 climate features in addition to "wet bulb". The train/val/test contains 28/10/10 months of data. (3) *Electricity* [4] contains the hourly electricity consumption of 321 clients in 2 years. The train/val/test contains 15/3/4 months of data.

**Implementation details**   We use the ADAM (Kingma & Ba, 2014) optimizer with a constant learning rate 1e-3 to train our model. The number of training epochs is set to 10. The hyper-parameter $\lambda$ in (13) is set to 1 for all experiments. All experiments are implemented with PyTorch (Paszke et al., 2019) and conducted on four NVIDIA TITAN RTX 24GB GPUs. For most experimental settings, SCformer contains 2 encoder layers and 1 decoder layer. Several models are selected to compare with SCformer, including three transformer-based models: Informer (Zhou et al., 2021), LogTrans (Li et al., 2019), Reformer (Kitaev et al., 2019), one RNN-based model: LSTM (Hochreiter & Schmidhuber, 1997) and one TCN-based model: TCN (Bai et al., 2018). To better explore the SCAttention's performance, we train our SCformer without the dual task to get SCformer[†].

**Evalution metrics**   On each prediction window, we use two metrics which are commonly used in prediction task to evaluate performances, i.e., Mean Square Error (MSE) and Mean Absolute Error (MAE). They are defined as follows: $\text{MSE} = \frac{1}{n}\sum_{i=1}^{n}(y_i - \hat{y}_i)^2$, $\text{MAE} = \frac{1}{n}\sum_{i=1}^{n}|y_i - \hat{y}_i|$, where $y_i$ is ground truth and $\hat{y}_i$ is prediction result. For multivariate prediction, the metrics can be calculated and averaged on each single variable.

### 4.2 MAIN RESULTS

**Univariate Time-series Forecasting**   In the univariate setting, both dimensions $d_x$ and $d_y$ of the input and output variables are 1, which means only a single dimension variable is utilized to predict itself. Following the setting in Zhou et al. (2021), we regard "oil temperature" in ETT-series datasets, "wet bulb" in Weather, and "MT_320" in Electricity as the single feature, respectively. The results are shown in Table 1. We observe that SCformer with dual task training achieves the best results in most cases. For example, under predict-48 settings, compared to previous state-of-the-art results, SCformer has achieved 49% ($0.108 \to 0.055$) relative improvement on MSE in ETTh1, 1% ($0.103 \to 0.102$) in ETTh2, 48% ($0.044 \to 0.023$) in ETTm1, 16% ($0.164 \to 0.137$) in Weather, and 16% ($0.311 \to 0.262$) in Electricity. Besides, SCformer outperforms SCformer[†] in most cases, which shows the effectiveness of dual task training. Further evaluation of the dual task is presented in the ablation study. Moreover, SCformer[†] can achieve better results than other models even without employing dual tasks, which illustrates SCAttention is more effective than other sparse-attention mechanisms, i.e., ProbSparse in Informer, LogSparse in LogTrans, and LSHSparse in Reformer. The overall performance of Transformer-based models is better than RNN-based model and TCN-based model, proving the potential of the Transformer-based model in time-series forecasting.

**Multivariate Time-series Forecasting**   Multivariate time-series forecasting aims at predicting multi-dimensional future values with multi-dimensional inputs. We can handle multivariate prediction by simply modifying the fully connected layer and embedding layer in SCformer. From Table 2, we find that the conclusion drawn in univariate prediction still holds in multivariate prediction. SCformer is also better than other sparse models, LSTM and TCN in multivariate prediction. For example, under predict-48 settings, compared to previous state-of-the-art results, SCformer or SCformer[†] has achieved 25% ($0.551 \to 0.413$) relative MSE improvement in ETTh1, 17% ($0.680 \to 0.563$) in ETTh2, 26% ($0.472 \to 0.351$) in ETTm1, 6% ($0.400 \to 0.377$) in Weather, and 13% ($0.267 \to 0.232$) in Electricity.

---

[2]https://github.com/zhouhaoyi/ETDstations.

[3]https://www.ncdc.noaa.gov/orders/qclcd/.

[4]https://archive.ics.uci.edu/ml/datasets/ElectricityLoadDiagrams20112014.

Table 1: Univariate time-series forecasting results on four datasets

| Models | | SCformer | | SCformer† | | Informer | | LogTrans | | Reformer | | LSTM | | TCN | |
|---|---|---|---|---|---|---|---|---|---|---|---|---|---|---|---|
| Metric | | MSE | MAE | MSE | MAE | MSE | MAE | MSE | MAE | MSE | MAE | MSE | MAE | MSE | MAE |
| ETTh1 | 24 | **0.036** | **0.148** | 0.047 | 0.170 | 0.062 | 0.178 | 0.059 | 0.191 | 0.172 | 0.319 | 0.094 | 0.232 | 0.079 | 0.213 |
| | 48 | **0.055** | **0.183** | 0.058 | 0.189 | 0.108 | 0.245 | 0.111 | 0.263 | 0.228 | 0.395 | 0.175 | 0.322 | 0.113 | 0.259 |
| | 168 | **0.081** | **0.222** | 0.087 | 0.233 | 0.146 | 0.294 | 0.155 | 0.309 | 1.460 | 1.089 | 0.210 | 0.352 | 0.173 | 0.338 |
| | 336 | **0.094** | **0.238** | 0.101 | 0.252 | 0.208 | 0.363 | 0.196 | 0.370 | 1.728 | 0.978 | 0.556 | 0.644 | 0.238 | 0.396 |
| | 720 | **0.121** | **0.276** | 0.146 | 0.306 | 0.193 | 0.365 | 0.217 | 0.379 | 1.948 | 1.226 | 0.635 | 0.704 | 0.246 | 0.399 |
| ETTh2 | 24 | 0.077 | 0.211 | **0.073** | **0.203** | 0.079 | 0.206 | 0.080 | 0.221 | 0.235 | 0.369 | 0.135 | 0.275 | 0.083 | 0.218 |
| | 48 | **0.102** | 0.249 | 0.107 | 0.258 | 0.103 | **0.240** | 0.107 | 0.262 | 0.434 | 0.505 | 0.172 | 0.318 | 0.111 | 0.257 |
| | 168 | **0.141** | 0.300 | 0.160 | 0.316 | 0.143 | **0.296** | 0.176 | 0.344 | 0.961 | 0.797 | 0.359 | 0.470 | 0.188 | 0.337 |
| | 336 | 0.179 | 0.335 | 0.189 | 0.344 | **0.171** | **0.327** | 0.175 | 0.345 | 1.532 | 1.060 | 0.516 | 0.548 | 0.230 | 0.387 |
| | 720 | **0.183** | 0.345 | 0.205 | 0.367 | 0.184 | **0.339** | 0.185 | 0.349 | 1.862 | 1.543 | 0.562 | 0.613 | 0.266 | 0.413 |
| ETTm1 | 24 | **0.013** | **0.085** | 0.014 | 0.091 | 0.051 | 0.153 | 0.061 | 0.192 | 0.055 | 0.170 | 0.099 | 0.201 | 0.032 | 0.139 |
| | 48 | **0.023** | **0.112** | 0.027 | 0.126 | 0.092 | 0.217 | 0.156 | 0.322 | 0.229 | 0.340 | 0.289 | 0.371 | 0.044 | 0.164 |
| | 96 | **0.041** | **0.152** | 0.046 | 0.165 | 0.119 | 0.249 | 0.229 | 0.397 | 0.854 | 0.675 | 0.255 | 0.370 | 0.071 | 0.202 |
| | 288 | **0.073** | **0.208** | 0.074 | **0.208** | 0.181 | 0.320 | 0.362 | 0.512 | 0.962 | 1.107 | 0.480 | 0.528 | 0.176 | 0.332 |
| | 672 | **0.097** | **0.244** | 0.105 | 0.249 | 0.204 | 0.345 | 0.450 | 0.582 | 1.605 | 1.312 | 0.988 | 0.805 | 0.213 | 0.355 |
| Weather | 24 | 0.092 | **0.207** | **0.088** | **0.207** | 0.107 | 0.223 | 0.120 | 0.247 | 0.197 | 0.329 | 0.107 | 0.222 | 0.112 | 0.239 |
| | 48 | 0.137 | 0.256 | **0.131** | **0.252** | 0.164 | 0.282 | 0.182 | 0.312 | 0.268 | 0.381 | 0.166 | 0.298 | 0.165 | 0.295 |
| | 168 | **0.214** | **0.337** | 0.220 | 0.338 | 0.226 | 0.338 | 0.267 | 0.387 | 0.590 | 0.552 | 0.305 | 0.404 | 0.233 | 0.363 |
| | 336 | **0.232** | **0.349** | 0.261 | 0.381 | 0.241 | 0.352 | 0.299 | 0.416 | 1.692 | 0.945 | 0.404 | 0.476 | 0.278 | 0.396 |
| | 720 | 0.286 | 0.389 | 0.297 | 0.411 | **0.259** | **0.367** | 0.274 | 0.387 | 1.887 | 1.352 | 0.784 | 0.709 | 0.425 | 0.491 |
| Electricity | 48 | **0.255** | **0.362** | 0.262 | 0.364 | 0.335 | 0.423 | 0.360 | 0.455 | 0.917 | 0.840 | 0.475 | 0.509 | 0.473 | 0.506 |
| | 168 | **0.262** | **0.371** | 0.276 | 0.380 | 0.408 | 0.466 | 0.410 | 0.481 | 1.635 | 1.515 | 0.703 | 0.617 | 0.311 | 0.411 |
| | 336 | **0.293** | **0.392** | 0.303 | 0.400 | 0.451 | 0.488 | 0.482 | 0.521 | 3.448 | 2.088 | 1.186 | 0.854 | 0.330 | 0.427 |
| | 720 | **0.318** | **0.423** | 0.337 | 0.438 | 0.466 | 0.499 | 0.522 | 0.551 | 4.745 | 3.913 | 1.473 | 0.910 | 0.414 | 0.484 |
| | 960 | **0.368** | **0.461** | 0.388 | 0.478 | 0.470 | 0.520 | 0.546 | 0.563 | 6.841 | 4.913 | 1.493 | 0.926 | 0.430 | 0.503 |

[1] Reported metrics in Informer, LogTrans, Reformer, LSTM come from (Zhou et al., 2021).

Table 2: Multivariate time-series forecasting results on four datasets

| Models | | SCformer | | SCformer† | | Informer | | LogTrans | | Reformer | | LSTM | | TCN | |
|---|---|---|---|---|---|---|---|---|---|---|---|---|---|---|---|
| Metric | | MSE | MAE | MSE | MAE | MSE | MAE | MSE | MAE | MSE | MAE | MSE | MAE | MSE | MAE |
| ETTh1 | 24 | **0.351** | **0.412** | 0.372 | 0.427 | 0.509 | 0.523 | 0.656 | 0.600 | 0.887 | 0.630 | 0.536 | 0.528 | 0.576 | 0.518 |
| | 48 | **0.413** | **0.451** | 0.430 | 0.457 | 0.551 | 0.563 | 0.670 | 0.611 | 1.159 | 0.750 | 0.616 | 0.577 | 0.572 | 0.543 |
| | 168 | **0.654** | **0.592** | 0.695 | 0.614 | 0.878 | 0.722 | 0.888 | 0.766 | 1.686 | 0.996 | 1.058 | 0.725 | 0.779 | 0.649 |
| | 336 | **0.769** | **0.672** | 0.834 | 0.689 | 0.884 | 0.753 | 0.942 | 0.792 | 1.919 | 1.090 | 1.152 | 0.794 | 1.189 | 0.846 |
| | 720 | 0.952 | **0.766** | 0.964 | 0.777 | **0.941** | 0.768 | 1.109 | 0.843 | 2.177 | 1.218 | 1.682 | 1.018 | 1.110 | 0.863 |
| ETTh2 | 24 | **0.250** | **0.345** | 0.310 | 0.402 | 0.446 | 0.523 | 0.726 | 0.638 | 1.381 | 1.475 | 1.049 | 0.689 | 0.406 | 0.494 |
| | 48 | **0.563** | **0.567** | 0.650 | 0.609 | 0.934 | 0.733 | 1.728 | 0.944 | 1.715 | 1.585 | 1.331 | 0.805 | 0.680 | 0.641 |
| | 168 | **1.327** | **0.894** | 1.338 | 0.906 | 1.512 | 0.996 | 3.944 | 1.573 | 4.484 | 1.650 | 3.987 | 1.560 | 2.661 | 1.312 |
| | 336 | **1.644** | **0.980** | 1.685 | 0.985 | 1.665 | 1.035 | 3.711 | 1.587 | 3.798 | 1.508 | 3.276 | 1.375 | 2.932 | 1.452 |
| | 720 | **2.123** | **1.189** | 2.247 | 1.205 | 2.340 | 1.209 | 2.817 | 1.356 | 5.111 | 1.793 | 3.711 | 1.520 | 3.595 | 1.558 |
| ETTm1 | 24 | 0.378 | 0.397 | 0.383 | **0.390** | **0.325** | 0.440 | 0.341 | 0.495 | 0.598 | 0.489 | 0.511 | 0.517 | 0.472 | 0.438 |
| | 48 | 0.355 | **0.400** | **0.351** | 0.416 | 0.472 | 0.537 | 0.495 | 0.527 | 0.952 | 0.645 | 1.280 | 0.819 | 0.568 | 0.484 |
| | 96 | 0.437 | 0.479 | **0.426** | **0.472** | 0.642 | 0.626 | 0.674 | 0.674 | 1.267 | 0.795 | 1.195 | 0.785 | 0.607 | 0.528 |
| | 288 | **0.466** | **0.487** | 0.499 | 0.494 | 1.219 | 0.871 | 1.728 | 1.656 | 1.632 | 0.886 | 1.598 | 0.952 | 0.670 | 0.611 |
| | 672 | **0.640** | **0.571** | 0.686 | 0.624 | 1.651 | 1.002 | 1.865 | 1.721 | 1.943 | 1.006 | 2.530 | 1.259 | 0.990 | 0.752 |
| Weather | 24 | **0.305** | **0.356** | 0.314 | 0.363 | 0.353 | 0.381 | 0.365 | 0.405 | 0.583 | 0.497 | 0.476 | 0.464 | 0.317 | 0.376 |
| | 48 | 0.381 | 0.422 | **0.377** | **0.420** | 0.464 | 0.455 | 0.496 | 0.485 | 0.633 | 0.556 | 0.763 | 0.589 | 0.400 | 0.439 |
| | 168 | **0.494** | **0.501** | 0.507 | 0.515 | 0.592 | 0.531 | 0.649 | 0.573 | 1.228 | 0.763 | 0.948 | 0.713 | 0.565 | 0.545 |
| | 336 | **0.529** | **0.537** | 0.554 | 0.543 | 0.623 | 0.546 | 0.666 | 0.584 | 1.770 | 0.997 | 1.497 | 0.889 | 0.610 | 0.577 |
| | 720 | **0.598** | **0.561** | 0.614 | 0.578 | 0.685 | 0.575 | 0.741 | 0.616 | 2.548 | 1.407 | 1.314 | 0.875 | 0.641 | 0.603 |
| Electricity | 48 | **0.232** | **0.344** | 0.249 | 0.359 | 0.269 | 0.351 | 0.267 | 0.366 | 1.312 | 0.911 | 0.388 | 0.444 | 0.293 | 0.386 |
| | 168 | 0.280 | 0.382 | **0.273** | **0.374** | 0.300 | 0.376 | 0.290 | 0.382 | 1.453 | 0.975 | 0.492 | 0.498 | 0.319 | 0.415 |
| | 336 | **0.294** | 0.393 | 0.310 | 0.406 | 0.311 | **0.385** | 0.305 | 0.395 | 1.507 | 0.978 | 0.778 | 0.629 | 0.398 | 0.479 |
| | 720 | **0.302** | **0.400** | 0.313 | 0.408 | 0.308 | 0.385 | 0.311 | 0.397 | 1.883 | 1.002 | 1.528 | 0.945 | 0.326 | 0.412 |
| | 960 | **0.319** | 0.417 | 0.323 | 0.415 | 0.328 | **0.406** | 0.333 | 0.413 | 1.973 | 1.185 | 1.343 | 0.886 | 0.374 | 0.456 |

[1] Reported metrics in Informer, LogTrans, Reformer, LSTM come from (Zhou et al., 2021).

## 4.3 ABLATION STUDY

**SCAttention vs. sparse-attention family.** We conduct experiments on the ETTh1 dataset with different settings to compare several different sparse attention mechanisms. For a fair comparison, we do not use the dual task training in all the experiments. The segment length of SCformer is set to 24. From Table 3, we observe that the performance increases as the encoder input becomes longer benefiting from more historical information. With longer inputs of the encoder, SCAttention achieves better performance than canonical self-attention (SCformer‡). Moreover, compared with other sparse mechanisms, SCformer† achieves the best performance, which further illustrates the effectiveness of the proposed SCAttention.

Table 3: Comparison of SCAttention and other sparse attention

| Prediction horizon | | 336 | | | 720 | | |
|---|---|---|---|---|---|---|---|
| Encoder's input length | | 336 | 720 | 1440 | 720 | 1440 | 2880 |
| SCformer[†] | MSE | 0.236 | 0.215 | 0.198 | 0.248 | 0.237 | 0.209 |
| | MAE | 0.480 | 0.414 | 0.377 | 0.506 | 0.383 | 0.375 |
| SCformer[‡] | MSE | 0.248 | 0.247 | - | 0.284 | - | - |
| | MAE | 0.469 | 0.452 | - | 0.539 | - | - |
| LogTrans | MSE | 0.256 | 0.233 | - | 0.264 | - | - |
| | MAE | 0.496 | 0.412 | - | 0.523 | - | - |
| Informer | MSE | 0.243 | 0.225 | 0.212 | 0.258 | 0.238 | 0.224 |
| | MAE | 0.487 | 0.404 | 0.381 | 0.503 | 0.399 | 0.387 |
| Reformer | MSE | 1.848 | 1.832 | 1.817 | 2.094 | 2.055 | 2.032 |
| | MAE | 1.054 | 1.027 | 1.010 | 1.363 | 1.306 | 1.334 |

[1] SCformer[‡] replaces SCAttention in SCformer[†] with canonical attention.
[2] The "-" indicates the out-of-memory (>32GB).
[3] Results except SCformer come from Zhou et al. (2021).

Table 4: Influence of segment length on different datasets

| Segment length | | 4 | 6 | 12 | 24 | 48 |
|---|---|---|---|---|---|---|
| ETTh1 | MSE | 0.100 | 0.109 | 0.102 | 0.094 | 0.106 |
| | MAE | 0.249 | 0.263 | 0.252 | 0.242 | 0.261 |
| ETTh2 | MSE | 0.208 | 0.205 | 0.202 | 0.184 | 0.195 |
| | MAE | 0.367 | 0.364 | 0.365 | 0.349 | 0.362 |
| ETTm1 | MSE | 0.069 | 0.061 | 0.067 | 0.086 | 0.066 |
| | MAE | 0.205 | 0.192 | 0.201 | 0.231 | 0.199 |
| Weather | MSE | 0.268 | 0.263 | 0.260 | 0.272 | 0.276 |
| | MAE | 0.383 | 0.380 | 0.376 | 0.393 | 0.391 |
| Electricity | MSE | 0.299 | 0.300 | 0.302 | 0.301 | 0.301 |
| | MAE | 0.398 | 0.399 | 0.396 | 0.396 | 0.401 |

Table 5: Ablation of the dual task

| Prediction type | | Univariate | | | | | Multivariate | | | | |
|---|---|---|---|---|---|---|---|---|---|---|---|
| Prediction horizon | | 24 | 48 | 168 | 336 | 720 | 24 | 48 | 168 | 336 | 720 |
| SCformer | MSE | 0.036 | 0.055 | 0.081 | 0.094 | 0.121 | 0.351 | 0.413 | 0.654 | 0.769 | 0.952 |
| | MAE | 0.148 | 0.183 | 0.222 | 0.238 | 0.276 | 0.412 | 0.451 | 0.592 | 0.672 | 0.766 |
| SCformer[†] | MSE | 0.047 | 0.058 | 0.087 | 0.101 | 0.146 | 0.372 | 0.430 | 0.695 | 0.834 | 0.964 |
| | MAE | 0.170 | 0.189 | 0.233 | 0.252 | 0.306 | 0.427 | 0.457 | 0.614 | 0.689 | 0.777 |
| Transformer | MSE | 0.055 | 0.066 | 0.095 | 0.109 | 0.170 | 0.427 | 0.446 | 0.818 | 0.876 | 1.011 |
| | MAE | 0.180 | 0.198 | 0.239 | 0.261 | 0.318 | 0.458 | 0.472 | 0.678 | 0.738 | 0.790 |
| Transformer[†] | MSE | 0.058 | 0.075 | 0.107 | 0.119 | 0.190 | 0.420 | 0.489 | 0.857 | 0.967 | 1.110 |
| | MAE | 0.187 | 0.220 | 0.258 | 0.272 | 0.349 | 0.462 | 0.511 | 0.694 | 0.762 | 0.842 |

[1] SCformer and Transformer is trained with dual task while SCformer[†] and Transformer[†] is trained without dual task.
[2] Transformer denotes a basic Transformer model which replaces SCAttention in SCformer with canonical attention.

**Analysis of the segment length.** The segment length $L_{seg}$ is a critical hyperparameter in SCAttention. We conduct experiments (input-336 to predict-336) on different datasets to explore the relationship between the segment length and time series. In order to eliminate the influence of other factors, all experiments are conducted without dual task training. As shown in Table 4, its optimal setting coincides with the implicit period of the time series.

**The effect of the dual task.** In order to explore the effect of the dual task, we train two models with or without the additional dual task to obtain 4 sets of model settings, i.e., SCformer with dual task (SCformer) and without dual task (SCformer[†]), vanilla Transformer with dual task (Transformer) and without dual task (Transformer[†]). To get a more comprehensive comparison, experiments are conducted on the ETTh1 dataset under different prediction types and horizons. From Table 5, we observe that the dual task training is useful for both these two models. The overall improvement ratios of MSE and MAE have reached 8.2% and 5.2% respectively.

## 5 CONCLUSION

This paper studies the long sequence time series forecasting problem which has urgent application demands. To tackle this problem, we propose a sparse and efficient attention mechanism called SCAttention which utilizes correlations between segment pairs to discover dependencies and aggregate information in time series. Under various experimental settings in different datasets, SCformer with SCAttention mechanism can yield state-of-the-art prediction performance, which means our SCAttention can extract the dependencies in time series in a better way than the previous sparse-attention family. Besides, to achieve better performance, we design a simple and effective dual task to train our model, which restores the past with the future. This dual task can be used not only in SCformer but also in other predictive models. Future exploration will focus on the combination of SCAttention and other sparse mechanisms.

ETHICS STATEMENT

This research does not involve any ethical issues. All the datasets in this paper are public for research use, without privacy leakage.

REPRODUCIBILITY STATEMENT

We preprocess all the datasets following the steps in an existing public repository: `https://github.com/zhouhaoyi/Informer2020`, which is the official PyTorch implementation of Informer (Zhou et al., 2021). We have reimplemented the original Transformer as `http://nlp.seas.harvard.edu/2018/04/03/attention.html` and modified some modules to get our SCformer model. Our code will be released soon if the paper is accepted.

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

## A  APPENDIX

### A.1  COMPARE WITH AUTO-CORRELATION IN AUTOFORMER

As shown in Table 6, we conduct experiments on ETTm2 dataset to compare our SCAttention with Auto-Correlation in Autoformer. We fix the input length as 96 and adopt the forecasting setting as multivariate. In order to compare these two attention mechanisms fairly, we train our SCformer model without the dual task and remove the progressive decomposition module in Autoformer. Compared with Auto-Correlation, our SCAttention can achieve comparable performance without complicated FFT calculations.

Table 6: Comparison of Auto-Correlation and SCAttention

| Encoder's input length | | 96 | | |
|---|---|---|---|---|
| Prediction horizon | | 336 | 720 | 1440 |
| SCAttention | MSE | **0.323** | 0.431 | **0.552** |
| | MAE | **0.370** | 0.425 | 0.511 |
| Auto-Correlation | MSE | 0.339 | **0.422** | 0.555 |
| | MAE | 0.372 | **0.419** | **0.496** |

## A.2 EFFECT OF HYPER-PARAMETER

To explore the effect of hyper-parameter, we conduct experiments about the number of heads and encoder layers on ETTh1 dataset. We adopt the forecasting setting as multivariate and predict-168. As shown in Table 7, setting the number of heads to 4 is the best. The number of decoder layers $M$ is fixed to 1, we study the effect of the number of encoder layers $N$ on the model. As shown in Table 8, when the number of encoder layers is small, the modeling and predictive ability of SCformer is insufficient. While the number of layers is large, the prediction performance is not significantly improved with the amount of parameters increasing. Therefore, it is best to set $N$ to 2.

Table 7: Effect of the number of heads in SCAttention

| n_heads | MSE | MAE |
|---|---|---|
| 4 | **0.654** | 0.592 |
| 5 | 0.671 | 0.603 |
| 6 | 0.684 | 0.611 |
| 7 | 0.668 | **0.590** |
| 8 | 0.661 | 0.600 |

Table 8: Effect of the number of encoder layers

| layer_num | MSE | MAE |
|---|---|---|
| 1 | 0.691 | 0.620 |
| 2 | **0.654** | 0.592 |
| 3 | 0.658 | **0.586** |
| 4 | 0.661 | 0.590 |

