# OpenReview forum: "SCformer: Segment Correlation Transformer for Long Sequence Time Series Forecasting"
_ICLR.cc/2022/Conference — ICLR 2022 Submitted_

### Official Review · Reviewer_5TYz · 2021-11-02

**Correctness:** 3
**Technical Novelty And Significance:** 2
**Empirical Novelty And Significance:** 3
**Recommendation:** 5
**Confidence:** 5

**Main Review:**

The paper studies the long time series sequence prediction problem with Transformer. The main contribution is proposing a SCAttention mechanism to replace the original attention operation by dividing the whole sequence into equal-length segments and calculating attention among them, which decreases the computation cost significantly.
Generally, the paper writing is clear, and the experimental results seem promising. However, I have the following concerns about this paper:
1. The methodology is quite intuitive, while it is reasonable to decrease the computation cost by segmenting the sequence, it is not well discussed why the performance could be improved significantly?
2. It is wired that the performance in Table 1 and Table 2 is different from that reported in the Informer paper. Is there any reason or did I misunderstand anything?
3. In the ablation study about SCAttention, is the SCformer with canonical attention the same as the original transformer? If yes, why the performance is so good compared to other transformer variants?
4. While experimental shown useful, the novelty of the dual reverse prediction task is limited. Similar ideas have been explored in other time series and traffic forecasting tasks, e.g., use the future 12 steps to predict the historical 12 steps, use bi-directional LSTMs.
5. finally, it would be better to analyze the model from more aspects, including the influence of encoder/decoder blocks, the number of parameters, analysis and reason about how to select the suitable segment length from Table 4, the variance of the performance, and so on.

Among these comments, 1, 2, and 3 are my key concerns. I may update the score if they can be adequately solved.

**Summary Of The Paper:**

The paper studies the long time series sequence prediction problem with Transformer. The main contribution is proposing a SCAttention mechanism to replace the original attention operation by dividing the whole sequence into equal-length segments and calculating attention among them, which decreases the computation cost significantly.

**Summary Of The Review:**

The technical contribution of this paper is marginal and the experimental analysis can be improved.

---

> ### Author Response · Authors · 2021-11-21
> **Response to Concern 1**
>
> We thank the reviewer for the thoughtful comments and constructive suggestions. Below we answer your questions:
>
> **Q1. The methodology is quite intuitive, while it is reasonable to decrease the computation cost by segmenting the sequence, it is not well discussed why the performance could be improved significantly?**
>
> We think there are two reasons. Firstly, for long sequence time-series forecasting, there are too many elements involved in the attention calculation in the ordinary Transformer, which will cause the attention score after softmax to be too small and reduce the correlation extraction ability. On the contrary, calculating the correlation between segments can greatly reduce the number of elements involved in the attention operation. Secondly, the locality in the time series is important. Periodic sequences show strong dependencies in the same period. However, the ordinary Transformer processes each element independently, and introduces position information through positional encoding. Our SCAttention keeps the local connection pattern between time steps in each segment, which is essential for the generation of the real time series.

---

> ### Author Response · Authors · 2021-11-21
> **Response to Concern 2**
>
> **Q2. It is wired that the performance in Table 1 and Table 2 is different from that reported in the Informer paper. Is there any reason or did I misunderstand anything?**
>
> The performance in Table 1 and Table 2 is consistent with the Informer paper (we directly compare with the results reported in the Informer paper). Please note that our tables are displayed vertically, while tables in the Informer paper are horizontal.

---

> > ### Comment · Reviewer_5TYz · 2021-11-29
> > **Still confused**
> >
> > Thanks for the response, I have double-checked your Table 1 with Table 1 in Zhou et.al. However, I still can not understand why your results are consistent. Both papers present results of the same method vertically and different methods horizontally.

---

> > > ### Author Response · Authors · 2021-11-29
> > > **Response to your confusion**
> > >
> > > I just found the Informer paper's authors have updated the experiment results of all methods due to the change in data scaling. But we are referring to the second version which is the version received by AAAI (i.e. https://arxiv.org/pdf/2012.07436v2.pdf), not the latest version (i.e. https://arxiv.org/pdf/2012.07436v3.pdf). Thanks for your reminder.

---

> ### Author Response · Authors · 2021-11-21
> **Response to Concern 3**
>
> **Q3. In the ablation study about SCAttention, is the SCformer with canonical attention the same as the original transformer? If yes, why the performance is so good compared to other transformer variants?}**
>
>
> Yes, the SCformer with canonical attention is similar to the original transformer. The results of other transformer variants come from Table 3 in the Informer paper. We just found that ablation experiments in Informer reduce setting as \{batch\_size=8, dim=64\} to isolate the memory efficient problem. We are sorry for this oversight. Thanks for your careful observation and reminder. We re-do the experiments under the same settings (batch\_size=8, dim=64). Results have been updated.

---

> ### Author Response · Authors · 2021-11-21
> **Response to Concern 4**
>
> **Q4. While experimental shown useful, the novelty of the dual reverse prediction task is limited. Similar ideas have been explored in other time series and traffic forecasting tasks, e.g., use the future 12 steps to predict the historical 12 steps, use bi-directional LSTMs.**
>
> To our best knowledge, though similar ideas have been used in some other fields such as machine translation [1] and image style transfer [2], we are the first to introduce it into long sequence time-series forecasting. Moreover, our dual prediction is essentially different from dual tasks. Dual tasks used in [1][2] actually perform an inverse transformation from the output to the input with a different network, while in this paper, we use the same prediction network for dual prediction. In fact, dual prediction can be viewed as an augmentation of data to enhance the same prediction network rather than performing a dual task. If you found that our dual prediction has existed in other time series and traffic forecasting tasks, please list relevant papers. Besides, bi-directional LSTMs where the intermediate node get information from the nodes in both directions can not be understood as an example of the dual task. According to the way you understand, Transformer itself can also be regarded as the dual task because it is bidirectional (see [3]), which is obviously very different from our dual task.
>
> [1] Edunov, S. , et al. "Understanding Back-Translation at Scale." Proceedings of the 2018 Conference on Empirical Methods in Natural Language Processing 2018.
>
> [2] Zhu, Jun-Yan, et al. "Unpaired image-to-image translation using cycle-consistent adversarial networks." Proceedings of the IEEE international conference on computer vision. 2017.
>
> [3] Devlin, Jacob, et al. "Bert: Pre-training of deep bidirectional transformers for language understanding." arXiv preprint arXiv:1810.04805 (2018).

---

> ### Author Response · Authors · 2021-11-21
> **Response to Concern 5**
>
> **Q5. Finally, it would be better to analyze the model from more aspects, including the influence of encoder/decoder blocks, the number of parameters, analysis and reason about how to select the suitable segment length from Table 4, the variance of the performance, and so on.**
>
> We conduct more comprehensive experiments including the number of encoder layers and heads. We have added these studies in Appendix A.2, Page 12 of the updated manuscript. To explore the effect of hyper-parameter, we conduct experiments about the number of heads and encoder layers on ETTh1 dataset. We adopt the forecasting setting as multivariate and predict-168. As shown in the Table, setting the number of heads to 4 is the best. The number of decoder layers $M$ is fixed to 1, we study the effect of the number of encoder layers $N$ on the model. When the number of encoder layers is small, the modeling and predictive ability of SCformer is insufficient. While the number of layers is large, the prediction performance is not significantly improved with the amount of parameters increasing. Therefore, it is best to set $N$ to 2.
>
> |n_heads | MSE | MAE |
> | :-----| :---- | :---- |
> | 4  | **0.654**  |  0.592  |
> | 5  | 0.671  |  0.603 |
> | 6  | 0.684  |  0.611 |
> | 7  | 0.668  |  **0.590** |
> | 8  | 0.661  |  0.600 |
>
>
> |layer_num | MSE | MAE |
> | :-----| :---- | :---- |
> | 1  | 0.691  |  0.620  |
> | 2  | **0.654**  |  0.592 |
> | 3  | 0.658  |  **0.586** |
> | 4  | 0.661  |  0.590 |
>
> As for how to select the suitable segment length, we perform ablation studies in Table 4 and select the segment length which can achieve best performance. Specifically, set the segment length to 24 for ETTh1 and ETTh2, 6 for ETTm1, 12 for Weather, 4 for Electricity. These selections well fit the prior knowledge of humans about these data. For example, the power consumption of transformers (ETTh1, ETTh2) takes the day as a cycle. The segment length of the household electricity consumption (Electricity) is 4 hours in line with intuition, because people usually work 4 hours in the morning and 4 hours in the afternoon. Four hours in the evening is the peak electricity consumption period. The segment length of Weather is 12 hours because of the large temperature difference between day and night.

---

### Official Review · Reviewer_pENX · 2021-11-02

**Correctness:** 2
**Technical Novelty And Significance:** 2
**Empirical Novelty And Significance:** 2
**Recommendation:** 3
**Confidence:** 4

**Main Review:**

Strengths:
1. The paper is well-motivated that the time complexity of the Transformer is quadratic to the number of time steps.
2. Using the dual task to improve the performance is interesting, and the experimental results show that the dual task could help to improve the results.
3. The experimenal results could demonstrate the effectiveness of the proposed segment correlation and dual task.

Weakness:
1. The novelty of the proposed segment correlation is limited, which simply truncates the long time series into small segments and calculates attention scores over the segments.
2. The time complexity $O(L^2//L^2_{seg})$ is incorrect, which should be $O(L^2//L_{seg})$ instead. $O(L^2//L^2_{seg})$ is the complexity among segments, and the complexity of each segment is not taken into consideration, which is $O(L_{seg})$
3. For the multivariate setting in table 5, Transformer trained without dual task performs better than Transformer with dual task. The results imply that the dual task might not be very useful to some degree.

**Summary Of The Paper:**

This paper introduces a SCFORMER, which replaces the canonical attention in the Transformer with the segment correlation attention. The motivation of using the segment correlation is to reduce the memory usage of the scale-dot product attention of the Transformer. To further improve the performance, the paper proposes a dual task, which use the current time series to predict the past time series.

**Summary Of The Review:**

This paper introduces a SCFORMER to address the problem that the complexity of the Transformer increases quadratically with the length of time series. The essential idea is to segment the input into small segments and calculate attention among the segments rather than each data point. To further improve the performance, this paper introduces an interesting dual task, which uses the current time series as input and will predict the historical time series. However, this the novelty of the segment correlation attention within the SCFORMER is limited and the time complexity is incorrect. Besides, the experimental results in table 5 do not sufficiently support the claim that the dual task could help model to achieve more robust results.

---

> ### Author Response · Authors · 2021-11-21
> **Response to Weakness 1**
>
> We thank the reviewer for the valuable comments. Below we answer your questions:
>
> **Q1. The novelty of the proposed segment correlation is limited, which simply truncates the long time series into small segments and calculates attention scores over the segments.**
>
> Our SCformer differs from Autoformer greatly in several aspects. 1. Autoformer leverages the series-level correlations, while our SCformer employs segment-level or subseries-level correlations. Specifically, we aggregate the segments by inter-segmentation similarities instead of aggregating the entire series by time delay similarities. 2. Instead of the complicated Fast Fourier Transforms calculation in Auto-Correlation used by Autoformer, we directly segment the time series based on implicit period and compute the correlation between segments. 3. We use the dot product between segments to calculate cross-correlation instead of auto-correlation in Auto-Correlation mechanism of Autoformer.
>
> Our Scformer is significant. 1. Since Scformer uses segment-level correlations instead of the whole series-level correlations, it can employ finer-grained information in time series while utilizing the strong local dependencies within the same period. 2. Compared with Autoformer, SCformer is much simpler and has better interprebility. 3. Our SCAttention can be combined with some other sparse mechanisms. For example, we can select segments in exponentially increasing intervals by combining LogSparse attention and SCAttention.

---

> ### Author Response · Authors · 2021-11-21
> **Response to Weakness 2**
>
> **Q2. The time complexity $O(L^2//L_{seg}^2)$ is incorrect, which should be $O(L^2//L_{seg})$ instead. $O(L^2//L_{seg}^2)$ is the complexity among segments, and the complexity of each segment is not taken into consideration, which is $O(L_{seg})$.**
>
> Thank you for discovering this. We simply regard the time complexity of the dot product between two vectors of length $L_{seg}$ as $O(1)$. But in fact, it is $O(L_{seg})$. Therefore, the time complexity of SCAttention should be $O(L^2//L_{seg})$ as you said. We have corrected this error.

---

> ### Author Response · Authors · 2021-11-21
> **Response to Weakness 3**
>
> **Q3. For the multivariate setting in table 5, Transformer trained without dual task performs better than Transformer with dual task. The results imply that the dual task might not be very useful to some degree.**
>
> Smaller MSE and MAE metrics mean better performance. In all cases except one, Transformer trained with dual task perform better than Transformer without dual task. Please check it again. The results show that the dual task is useful for prediction.

---

### Official Review · Reviewer_dkGP · 2021-11-08

**Correctness:** 3
**Technical Novelty And Significance:** 2
**Empirical Novelty And Significance:** 2
**Recommendation:** 3
**Confidence:** 4

**Details Of Ethics Concerns:**

N.A.

**Main Review:**


Strengths

* This paper is well organized.
* Long sequence time series forecasting is an interesting problem to investigate.
* The proposed SCformer is technically reasonable.

Weaknesses
* The overall technical novelty is limited.
* Several relevant works are not mentioned or compared.
* Some details of the proposed SCformer are not clear.

The major concern is over the technical novelty. The key idea of SCformer is very similar to Autoformer, both leverage the series-level correlations. In fact, Autoformer could be easily adapted to capture segment-level correlations.  In addition, the dual-task has been used in many previous works.

Another concern is that Autoformer and several related works are not compared:

[1] "Think globally, act locally: A deep neural network approach to high-dimensional time series forecasting." Sen, Rajat, Hsiang-Fu Yu, and Inderjit S. Dhillon NeurIPS 2019.
[2] "Modeling long-and short-term temporal patterns with deep neural networks." Lai, Guokun, Wei-Cheng Chang, Yiming Yang, and Hanxiao Liu, SIGIR 2018.
[3] "Shape and time distortion loss for training deep time series forecasting models." Vincent, L. E., and Nicolas Thome NeurIPS 2019.

The authors mentioned that their methods differ from Autoformer in the way of correlation computation and
aggregation. It is not clear which one is more efficient in practice.

How to determine segment length in the experiment?

In Figure 1, it is not clear why the input of decoder embedding is X_t1:X_t0

**Summary Of The Paper:**

This paper presents a Transformer-based model called SCformer to perform long sequence time series forecasting. The key idea is to replace the canonical self-attention with efficient segment correlation attention (SCAttention) mechanism to capture long short-term dependencies. Experiment results on several datasets showed the effectiveness of the proposed method.

**Summary Of The Review:**

See above

---

> ### Author Response · Authors · 2021-11-21
> **Response to Weakness 1**
>
> We thank the reviewer for the valuable comments. Below we address your questions:
>
> **Q1. The major concern is over the technical novelty. The key idea of SCformer is very similar to Autoformer, both leverage the series-level correlations. In fact, Autoformer could be easily adapted to capture segment-level correlations. In addition, the dual-task has been used in many previous works.**
>
> Our SCformer differs from Autoformer greatly in several aspects. 1. Autoformer leverages the series-level correlations, while our SCformer employs segment-level or subseries-level correlations. Specifically, we aggregate the segments by inter-segmentation similarities instead of aggregating the entire series by time delay similarities. 2. Instead of the complicated Fast Fourier Transforms calculation in Auto-Correlation used by Autoformer, we directly segment the time series based on implicit period and compute the correlation between segments. 3. We use the dot product between segments to calculate cross-correlation instead of auto-correlation in Auto-Correlation mechanism of Autoformer.
>
> Our SCformer is significant. 1. Since Scformer uses segment-level correlations instead of the whole series-level correlations, it can employ finer-grained information in time series while utilizing the strong local dependencies within the same period. 2. Compared with Autoformer, SCformer is much simpler and has better interprebility. 3. Our SCAttention can be combined with some other sparse mechanisms. For example, we can select segments in exponentially increasing intervals by combining LogSparse attention and SCAttention.
>
> For the dual prediction, to our best knowledge, though similar ideas have been used in some other fields such as machine translation [1] and image style transfer [2], we are the first to introduce it into long sequence time-series forecasting. Moreover, our dual prediction is essentially different from dual tasks. Dual tasks used in [1][2] actually perform an inverse transformation from the output to the input with a different network, while in this paper, we use the same prediction network for dual prediction. In fact, dual prediction can be viewed as an augmentation of data to enhance the same prediction network rather than performing a dual task. If you found that our dual prediction has existed in other time series and traffic forecasting tasks, please list relevant papers.
>
> [1] Edunov, S. , et al. "Understanding Back-Translation at Scale." Proceedings of the 2018 Conference on Empirical Methods in Natural Language Processing 2018.
>
> [2] Zhu, Jun-Yan, et al. "Unpaired image-to-image translation using cycle-consistent adversarial networks." Proceedings of the IEEE international conference on computer vision. 2017.

---

> ### Author Response · Authors · 2021-11-21
> **Response to Weakness 2**
>
> **Q2. Another concern is that Autoformer and several related works are not compared: [1] "Think globally, act locally: A deep neural network approach to high-dimensional time series forecasting." Sen, Rajat, Hsiang-Fu Yu, and Inderjit S. Dhillon NeurIPS 2019. [2] "Modeling long-and short-term temporal patterns with deep neural networks." Lai, Guokun, Wei-Cheng Chang, Yiming Yang, and Hanxiao Liu, SIGIR 2018. [3] "Shape and time distortion loss for training deep time series forecasting models." Vincent, L. E., and Nicolas Thome NeurIPS 2019.**
>
> We mainly compare with other Transformer-based models in this paper. We think non-Transformer-based models are equally interesting and thank you for pointing this out. The reason why Autoformer is not compared is that the code of Autoformer was not released before the ICLR submission deadline. The experimental settings of Autoformer such as prediction length and datasets in the previous version is different from the Informer paper and our paper. And the latest version of the Autoformer paper including more comprehensive results was updated recently. We add the results of comparison with Autoformer and LSTNet (the paper [2] you provided) as shown in the table. In order to be consistent with Autoformer, we conduct experiments for longer average prediction lengths with SCformer. The results of Autoformer and LSTNet come from the Autoformer paper.
>
> | Models | SCformer | Autoformer | LSTNet
> | -----: | :---- | :---- | :----
> | | MSE MAE | MSE MAE | MSE MAE
> | ETTh1  24| **0.351 0.412**  |  0.384 0.425  |  1.293 0.901
> | 48| 0.413 0.451  |  **0.392 0.419**  |  1.456 0.960
> | 168| 0.654   0.592| **0.490  0.481**|1.997  1.214
> | 336| 0.769   0.672|**0.505  0.484**|2.655  1.369
> | 720| 0.952   0.766|**0.498  0.500**|2.143  1.380
> | ETTh2  24| **0.250 0.345**| 0.261 0.341| 2.742  1.457
> | 48| 0.563   0.567|**0.312  0.373**|3.567  1.687
> | 168| 1.327   0.894|**0.457  0.455**|3.242  2.513
> | 336| 1.644   0.980|**0.471  0.475**|2.544  2.591
> | 720| 2.123   1.189|**0.474  0.484**|4.625  3.709
> | ETTm1  24| **0.378  0.397**|0.383   0.403|1.968  1.170
> | 48|**0.355  0.400**|0.454   0.453|1.999  1.215
> | 96| **0.437   0.479**|0.481   0.463|2.762  1.542
> | 288| **0.466  0.487**|0.634   0.528|1.257  2.076
> | 672| 0.640   0.571|0.606  0.542|1.917  2.941
>
> Though Autoformer outperforms our SCformer, time series decomposition which is very helpful for long sequence time-series forecasting is used in Autoformer but not in SCformer. The performance of Autoformer benefit greatly from the decomposition module, which has been confirmed in the ablation experiments of Autoformer: "With our proposed progressive decomposition architecture, prediction
> models can gain consistent promotion, especially as the prediction length increases". Therefore, we also do experiments on comparison of Auto-Correlation and SCAttention, which time series decomposition module in Autoformer is not included. Our ablation experimental settings are the same as the Autoformer paper. We have added these ablation studies in Appendix A.1, Page 12 of the updated manuscript.
>
> | pred length | 336 | 720 | 1440 |
> | :----- | :---- | :---- | :---- |
> | | MSE MAE | MSE MAE | MSE MAE |
> | SCAttention |**0.323 0.370**|0.431 0.425|**0.552** 0.511|
> | Auto-Correlation |0.339 0.372|**0.422,0.419**|0.555 **0.496**|
>
> The paper[1] utilize TCN-based models and it is not designed for long sequence time-series forecasting. In our paper, we have compared with the basic TCN model which does not work well for long sequence time-series forecasting. The innovation of paper [3] is that it introduces a new loss for prediction which can be also used in SCformer. There is no structural innovation in paper [3], so there is no need to compare.

---

> ### Author Response · Authors · 2021-11-21
> **Response to Weakness 3**
>
> **Q3. How to determine segment length in the experiment?**
>
> Experiments are needed to determine segment length, but divisor of the series' period is usually a good choice. For example, for the Electricity dataset which is the hourly electricity consumption obviously presents a daily pattern. It's a good choice to set segment length as 4 or 6 or 12 or 24 if we regard 24 as the period. In this paper, we conduct ablation experiments on segment length to determine optimal segment length as shown in Table 4. The optimal segment length of different datasets will be different according to the attributes of the dataset itself. For example, the power consumption of transformers (ETTh1, ETTh2) takes the day as a cycle. The segment length of the household electricity consumption (Electricity) is 4 hours in line with intuition, because people usually work 4 hours in the morning and 4 hours in the afternoon. Four hours in the evening is the peak electricity consumption period. The segment length of Weather is 12 hours because of the large temperature difference between day and night.
>
>
> **Q4. In Figure 1, it is not clear why the input of decoder embedding is X\_t1 : X\_t0.**
>
> Thank you for pointing out this mistake. $X_{t_1 : t_0}$ in Figure 1 should be $X_{t_d : t_0}$. See the decoder section for the definition of $X_{t_d}$ : "The inputs $X_{t_d : t_0}$ of the decoder are the latter parts of the inputs of the encoder, where $t_d$ is the start time for the inputs of the decoder and $1 \leq t_d \leq t_0$." We have corrected this error.

---

### Official Review · Reviewer_SdzH · 2021-11-23

**Correctness:** 3
**Technical Novelty And Significance:** 2
**Empirical Novelty And Significance:** 2
**Recommendation:** 5
**Confidence:** 4

**Main Review:**

Strengths:

- Exploiting periodical information for long time series forecasting is interesting
- Paper is fluent and easy to follow.

Weaknesses:
- Comparisons to earlier work
- Misses ablation study
- Limited novelty, dual prediction already exists in bidirectional RNN (LSTM, GRU) time series forecasting and it is not clear how the segmentation is computed, and how it is different than Wu et al 2021 (see below).

Concatenating the segmented time series representations corresponds to concatenating slightly different versions of a short sequence. It can be interesting to see if using the whole sequence is better than using the last segmented sequence. It misses the ablation study of using the longer sequence compared to using the last segmented sequence.

It misses the comparison to the earlier work on periodical time series forecasting:
- Cinar, Y. G., Mirisaee, H., Goswami, P., Gaussier, E., & Aït-Bachir, A. (2018). Period-aware content attention RNNs for time series forecasting with missing values. Neurocomputing, 312, 177-186.
- Wu, H., Xu, J., Wang, J., & Long, M. (2021, May). Autoformer: Decomposition transformers with auto-correlation for long-term series forecasting. In Thirty-Fifth Conference on Neural Information Processing Systems.

---

- Providing the equation or the algorithm for the SEG function in Eq. 6 can improve the reproducibility of the approach, and help to better understand the contribution of the paper.
- The best scores are presented in bold in Table 1 and 2, but not in Table 3, 4, and 5. Can you please show the best scores in bold in Table 3, 4, and 5?


**Summary Of The Paper:**

This paper studies long time series forecasting by using segment correlation attention. For this, time series is segmented into periodical shorter sequences, and attention is computed on each segment and final representation is formed by concatenating segments’ self-attention representations.

**Summary Of The Review:**

Long time series forecasting using periodical information is interesting. The novelty of the paper is limited as the dual task already exists in bi-directional RNN based models for time series. It misses ablation study of using only the last segmentation instead of the long time series.

---

> ### Author Response · Authors · 2021-11-24
> **Response to Weakness 1**
>
> We thank the reviewer for the valuable comments. Below we address your questions:
>
> **Q1. It misses the comparison to the earlier work on periodical time series forecasting:
> [1]Cinar, Y. G., Mirisaee, H., Goswami, P., Gaussier, E., & Aït-Bachir, A. (2018). Period-aware content attention RNNs for time series forecasting with missing values. Neurocomputing, 312, 177-186.
> [2]Wu, H., Xu, J., Wang, J., & Long, M. (2021, May). Autoformer: Decomposition transformers with auto-correlation for long-term series forecasting. In Thirty-Fifth Conference on Neural Information Processing Systems.**
>
> We mainly compare with other Transformer-based models in this paper. We think non-Transformer-based models are equally interesting and thank you for pointing this out. The reason why Autoformer is not compared is that the code of Autoformer was not released before the ICLR submission deadline. The experimental settings of Autoformer such as prediction length and datasets in the previous version is different from the Informer paper and our paper. And the latest version of the Autoformer paper including more comprehensive results was updated recently. We add the results of comparison with Autoformer and LSTNet which is an LSTM-based model as shown in the table. In order to be consistent with Autoformer, we conduct experiments for longer average prediction lengths with SCformer. The results of Autoformer and LSTNet come from the Autoformer paper.
>
> | Models | SCformer | Autoformer | LSTNet
> | -----: | :---- | :---- | :----
> | | MSE MAE | MSE MAE | MSE MAE
> | ETTh1  24| **0.351 0.412**  |  0.384 0.425  |  1.293 0.901
> | 48| 0.413 0.451  |  **0.392 0.419**  |  1.456 0.960
> | 168| 0.654   0.592| **0.490  0.481**|1.997  1.214
> | 336| 0.769   0.672|**0.505  0.484**|2.655  1.369
> | 720| 0.952   0.766|**0.498  0.500**|2.143  1.380
> | ETTh2  24| **0.250 0.345**| 0.261 0.341| 2.742  1.457
> | 48| 0.563   0.567|**0.312  0.373**|3.567  1.687
> | 168| 1.327   0.894|**0.457  0.455**|3.242  2.513
> | 336| 1.644   0.980|**0.471  0.475**|2.544  2.591
> | 720| 2.123   1.189|**0.474  0.484**|4.625  3.709
> | ETTm1  24| **0.378  0.397**|0.383   0.403|1.968  1.170
> | 48|**0.355  0.400**|0.454   0.453|1.999  1.215
> | 96| **0.437   0.479**|0.481   0.463|2.762  1.542
> | 288| **0.466  0.487**|0.634   0.528|1.257  2.076
> | 672| 0.640   0.571|0.606  0.542|1.917  2.941
>
> Though Autoformer outperforms our SCformer, time series decomposition which is very helpful for long sequence time-series forecasting is used in Autoformer but not in SCformer. The performance of Autoformer benefit greatly from the decomposition module, which has been confirmed in the ablation experiments of Autoformer: "With our proposed progressive decomposition architecture, prediction
> models can gain consistent promotion, especially as the prediction length increases". Therefore, we also do experiments on the comparison of Auto-Correlation and SCAttention, in which the time series decomposition module in Autoformer is not included. Our ablation experimental settings are the same as the Autoformer paper. We have added these ablation studies in Appendix A.1, Page 12 of the updated manuscript.
>
> | pred length | 336 | 720 | 1440 |
> | :----- | :---- | :---- | :---- |
> | | MSE MAE | MSE MAE | MSE MAE |
> | SCAttention |**0.323 0.370**|0.431 0.425|**0.552** 0.511|
> | Auto-Correlation |0.339 0.372|**0.422,0.419**|0.555 **0.496**|
>
> The paper[1] utilizes an LSTM-based model and it is not designed for long sequence time-series forecasting. It aims to predict the future with past missing values. The key problem we want to address is different.

---

> ### Author Response · Authors · 2021-11-24
> **Response to Weakness 3**
>
> **Q3. Limited novelty, dual prediction already exists in bidirectional RNN (LSTM, GRU) time series forecasting and it is not clear how the segmentation is computed, and how it is different than Wu et al 2021 (see below).**
>
> Our SCformer differs from Autoformer greatly in several aspects. 1. Autoformer leverages the series-level correlations, while our SCformer employs segment-level or subseries-level correlations. Specifically, we aggregate the segments by inter-segmentation similarities instead of aggregating the entire series by time delay similarities. 2. Instead of the complicated Fast Fourier Transforms calculation in Auto-Correlation used by Autoformer, we directly segment the time series based on implicit period and compute the correlation between segments. 3. We use the dot product between segments to calculate cross-correlation instead of auto-correlation in Auto-Correlation mechanism of Autoformer.
>
> Our SCformer is significant. 1. Since Scformer uses segment-level correlations instead of the whole series-level correlations, it can employ finer-grained information in time series while utilizing the strong local dependencies within the same period. 2. Compared with Autoformer, SCformer is much simpler and has better interprebility. 3. Our SCAttention can be combined with some other sparse mechanisms. For example, we can select segments in exponentially increasing intervals by combining LogSparse attention and SCAttention.
>
> For the dual prediction, to our best knowledge, though similar ideas have been used in some other fields such as machine translation [1] and image style transfer [2], we are the first to introduce it into long sequence time-series forecasting. Moreover, our dual prediction is essentially different from dual tasks. Dual tasks used in [1][2] actually perform an inverse transformation from the output to the input with a different network, while in this paper, we use the same prediction network for dual prediction. In fact, dual prediction can be viewed as an augmentation of data to enhance the same prediction network rather than performing a dual task. If you found that our dual prediction has existed in other time series and traffic forecasting tasks, please list relevant papers.
>
> [1] Edunov, S. , et al. "Understanding Back-Translation at Scale." Proceedings of the 2018 Conference on Empirical Methods in Natural Language Processing 2018.
>
> [2] Zhu, Jun-Yan, et al. "Unpaired image-to-image translation using cycle-consistent adversarial networks." Proceedings of the IEEE international conference on computer vision. 2017.

---

> ### Author Response · Authors · 2021-11-24
> **Response to Weakness 2**
>
> **Q2. Concatenating the segmented time series representations corresponds to concatenating slightly different versions of a short sequence. It can be interesting to see if using the whole sequence is better than using the last segmented sequence. It misses the ablation study of using the longer sequence compared to using the last segmented sequence.**
>
> I'm sorry that I didn’t understand what you mean. What does using the last segmented sequence mean?
>
> **Q3. Providing the equation or the algorithm for the SEG function in Eq. 6 can improve the reproducibility of the approach, and help to better understand the contribution of the paper.
> The best scores are presented in bold in Table 1 and 2, but not in Table 3, 4, and 5. Can you please show the best scores in bold in Table 3, 4, and 5?**
>
> The paper can't be updated today. We will address these problems in the next version of our paper. Thanks for your constructive suggestions.

---

### Decision · Program_Chairs · 2022-01-20

**Decision:**

Reject

**Comment:**

The paper proposes a Transformer-based model called SCformer to perform long sequence time series forecasting by computing efficient segment correlation attention. The reviewers think the method lacks novelty and the experiments need a detailed ablation study.